# Hydration/Dehydration Behavior of Hydroxyethyl Cellulose Ether in Aqueous Solution

**DOI:** 10.3390/molecules25204726

**Published:** 2020-10-15

**Authors:** Kengo Arai, Toshiyuki Shikata

**Affiliations:** 1Cellulose Research Unit, Tokyo University of Agriculture and Technology, 3-5-8 Saiwai-cho, Fuchu, Tokyo 183-8509, Japan; 1992.ken.arai@gmail.com; 2Division of Natural Resources and Eco-materials, Graduate School of Agriculture, Tokyo University of Agriculture and Technology, 3-5-8 Saiwai-cho, Fuchu, Tokyo 183-8509, Japan

**Keywords:** chemically modified cellulose ether, hydroxyethyl cellulose, methyl cellulose, hydration, dehydration, dielectric spectroscopy, relaxation time

## Abstract

Hydroxyethyl cellulose (HeC) maintains high water solubility over a wide temperature range even in a high temperature region where other nonionic chemically modified cellulose ethers, such as methyl cellulose (MC) and hydroxypropylmethyl cellulose (HpMC), demonstrate cloud points. In order to clarify the reason for the high solubility of HeC, the temperature dependence of the hydration number per glucopyranose unit, *n*_H_, for the HeC samples was examined by using extremely high frequency dielectric spectrum measuring techniques up to 50 GHz over a temperature range from 10 to 70 °C. HeC samples with a molar substitution number (*MS*) per glucopyranose unit by hydroxyethyl groups ranging from 1.3 to 3.6 were examined in this study. All HeC samples dissolve into water over the examined temperature range and did not show their cloud points. The value of *n*_H_ for the HeC sample possessing the *MS* of 1.3 was 14 at 20 °C and decreased gently with increasing temperature and declined to 10 at 70 °C. The *n*_H_ values of the HeC samples are substantially larger than the minimum critical *n*_H_ value of ca. 5 necessary to be dissolved into water for cellulose ethers such as MC and HpMC, even in a high temperature range. Then, the HeC molecules possess water solubility over the wide temperature range. The temperature dependence of *n*_H_ for the HeC samples and triethyleneglycol, which is a model compound for substitution groups of HeC, is gentle and they are similar to each other. This observation strongly suggests that the hydration/dehydration behavior of the HeC samples was essentially controlled by that of their substitution groups.

## 1. Introduction

Cellulose is the most abundant natural organic resource on the globe. Cellulose is a high-molecular weight polysaccharide which consists of β-1,4-d-glucopyranose units [1]. Native cellulose is insoluble in most usual solvents including water, due to its highly developed inter- and intra-molecular hydrogen bonding between hydroxy groups [2,3]. To improve this insolubility and to make use of cellulose for a wide range of applications, many kinds of chemically modified cellulose derivatives have been synthesized from natural cellulose. A series of chemically modified celluloses, such as water-soluble nonionic methyl, hydroxypropyl, hydroxyethyl and hydroxypropylmethyl cellulose ethers, anionic sodium carboxymethyl cellulose ether, and organic solvent-soluble ethyl and cyanoethyl cellulose ethers, and cellulose nitrate and cellulose acetate, have been developed by several chemical companies [1]. These chemically modified cellulose samples have been widely used in various areas because they exhibit useful properties, such as viscosity thickening, surfactant activity, film formation, adhesion, and so on [1,4,5,6,7].

Hydroxyethyl cellulose (HeC) is one of the water-soluble cellulose ethers, which has been commercially produced by the chemical reaction between cellulose and ethylene oxide. HeC molecules are composed of glucopyranose units randomly substituted by hydroxyethyl groups at the position of 2, 3, or 6, of which side chains can be monomers, dimers, or trimers [8]. Scheme 1 shows the chemical structure of HeC examined in this study. Due to its high water solubility, HeC is widely used for a variety of applications such as cosmetics [9], building materials [10], film forming materials [11,12], and pharmaceutical productions (drug delivery system) [13,14].

Although the methyl cellulose (MC) and hydroxypropyl methyl cellulose (HpMC) samples, which are the most prevalent two nonionic cellulose ethers, show high water solubility below room temperature, they lose their solubility and sometimes become turbid gels at higher temperatures, i.e., lower critical solution temperatures (LCSTs) [15,16,17]. On the other hand, HeC shows high solubility in water over a wide temperature range even at higher temperatures than 50 °C [18]. Giudice et al. [19] have investigated the viscoelastic properties of aqueous HeC solution by using various rheological techniques over a wide concentration and frequency range. Koda et al. [20] have compared the water retention capacity among MC, HeC, and hydroxypropyl cellulose (HpC) by using the compression method and differential scanning calorimetry measurements and concluded that the capacity of hydroxyethyl cellulose is the strongest. Ouaer et al. [21] have determined the intrinsic viscosity and the critical overlap concentration of HeC molecules in aqueous solution. However, these studies were carried out at room temperature ca. 25 or 0 °C and very few studies have reported on the influence of temperature on the molecular properties of HeC in aqueous solution. Moreover, few studies have focused on the reason why HeC molecules keep their high water solubility even at high temperatures, and the reason still remains unsolved. Water solubility, such a fundamental property, is essential to understand the physicochemical characteristics of HeC molecules possessing a higher potential to be used in broader and more effective applications than they are currently used.

To understand the water solubility of polymeric materials, the determination of hydration numbers for the materials is important. Although Arfin et al. [22] have investigated the hydration behavior of HeC in aqueous solution by using differential scanning calorimetry and Raman spectroscopy measurements, they did not discuss the hydration number and its temperature dependence of HeC samples.

Recently, our group developed a useful technique to determine the hydration numbers of solute molecules dissolved in water as functions of temperature using dielectric spectroscopic (DS) measurements performed in an extremely high frequency range up to 50 GHz (3.14 × 10^11^ s^−1^ in angular frequency (*ω*)) [23,24,25,26]. Because the relaxation strength of free water molecules can be precisely evaluated in this high frequency range, the amount of water molecules hydrated to solute molecules can be exactly determined. In a previous study, the hydration numbers of poly(vinyl alcohol) (PVA) and partially chemically modified PVA, which showed much higher solubility in water than PVA, were determined as functions of temperature by using DS techniques. The obtained results have revealed that the hydration number of chemically modified PVA is larger than that of PVA over all temperature ranges in the study [23]. Furthermore, the temperature dependencies of hydration numbers per glucopyranose unit for chemically modified cellulose, MC, HpMC, and hydroxyethylmethyl cellulose (HeMC), were also obtained by using DS techniques. The results have shown that additional substitution to MC by hydroxypropyl or hydroxyethyl groups is quite effective to extend solubility into water because the dehydration behavior of HpMC and HeMC with increasing temperature is much gentler than that of MC [24]. Therefore, the hydration number and its temperature dependence determined by DS measurements should be a key parameter to understand the water solubility of the materials. It is important to understand why chemically modified cellulose samples possess rather different temperature dependencies of hydration behavior and solubility, highly depending on the species of substitution groups for their many advanced practical applications, especially at high temperatures.

In this study, we report the temperature dependence of hydration number per glucopyranose unit, *n*_H_, for the HeC samples in aqueous solution over a wide temperature range from 10 to 70 °C, determined by using extremely high-frequency DS techniques. The substitution group dependence of *n*_H_ with increasing temperature is fully discussed by comparing the *n*_H_ values of HeC, MC, HpMC, and HeMC. Additionally, the molar substitution number (*MS*) dependence of the *n*_H_ values is also discussed using HeC samples with different *MS* quantities per glucopyranose unit by hydroxyethyl groups ranging from 1.3 to 3.6. The HeC samples examined in this study were coded as HeC(*he*:*M*_w_/10^3^) with numerical quantities, *he* and *M*_w_, which represent the molar substitution number per glucopyranose unit, *MS*, by hydroxyethyl groups, and the weight-average molecular weight. Finally, the reason for the high-water solubility of HeC molecules over a wide temperature range will be discussed.

## 2. Results and Discussion

### 2.1. Dielectric Behavior for Aqueous Solution of Hydroxyethyl Cellulose

Dielectric spectra (*ε*′ and *ε*″ vs. *ω*) for the aqueous solution of HeC(1.3:90) at *c* = 0.23 mol L^−^^1^(M) and *T* = 20 °C are shown in Figure 1a as a typical example. The dependence of *ε*′ and *ε*″ on *ω* was decomposed into five dielectric relaxation modes described by the Debye-type relaxation modes as follows:(1)$$\varepsilon^{\prime }=\sum_{j=1}^{5}\frac{\varepsilon_{j}}{1+\omega^{2}\tau_{j}^{2}}+\varepsilon_{\infty },\;\varepsilon^{\prime \prime }=\sum_{j=1}^{5}\frac{\varepsilon_{j}\omega \tau_{j}}{1+\omega^{2}\tau_{j}^{2}}$$
where *ε_j_*, *τ_j_*, and *ε*_∞_ represent the relaxation strength and time for a mode *j* and the high-frequency limiting electric permittivity, respectively. The broken lines drawn in the figure show the constituent Debye-type relaxation mode *ε_j_*′ and *ε_j_*″ (*j* = 1 to 5 from the fastest relaxation mode due to the shortest relaxation time). To verify the validity of decomposition into five Debye-type relaxation modes for the spectra, the residuals of *ε*′–*ε*_1_′–*ε_∞_* (red) and *ε*″–*ε*_1_″ (blue) are also plotted in the same figure, which obviously demonstrate the presence of modes *j* = 2, 3, 4, and 5. The *G*_DC_ values for the examined samples were lower than 400 μS.

A variety of useful methods were proposed to investigate the hydration behavior of solute molecules including large biomacromolecules like proteins and DNAs, such as extended depolarized light scattering (EDLS) [27,28,29,30], nuclear magnetic resonance (NMR) [31,32], neutron scattering (NS) [33], time-resolved fluorescence decay [34] and dielectric spectroscopic (DS) [23,26,30,35,36,37,38] measurements, and molecular dynamics simulation (MD) [39,40,41] techniques. The DS measurement is one of the most useful methods to explore hydration behavior because it can directly detect the dynamics of water molecules quite precisely [23,26,30,35,36,37,38]. In the case of DS measurements conducted in an extremely high frequency range up to 50 GHz, it has been well-known that dielectric spectra of the collective rotational relaxation process for water molecules are well described, with a single Debye-type mode at the relaxation time of *τ*_w_ = 9.0 ps (at *T* = 20 °C) in pure liquid water [42]. Dielectric spectra for aqueous solutions of solute molecules are precisely described as the summation of Debye-type relaxation mode of free water molecules, additional Debye-type relaxation modes assigned to the rotational relaxation process of hydrated water molecules, and that of dipoles attached to solute molecules in aqueous systems [23,26,30,35,36,37,38]. Then, we used the summation of Debye-type modes given by Equation (1) to describe the dielectric spectra of aqueous solutions examined in this study.

It has been well-known that the Cole−Cole [43] and Cole−Davidson [44] type relaxation functions are also useful to describe dielectric behavior showing non-single Debye-type dielectric spectra. Figure 1b shows a so-called Cole−Cole plot for the data represented in Figure 1a together with the best fit single Debye-type behavior shown by a solid semicircle of major dielectric dispersion observed at *ω* ~ 10^11^ s^−1^. Non-negligible deviation from the solid circle is clearly observed in data points in a lower frequency region, which represents the additional contribution of the modes *j* = 3, 4, and 5. Although the agreement between the solid semicircle and data points in the major dispersion portion seems reasonably well, the Cole−Cole or Cole−Davidson models are not the best methods to express such DS spectra, showing a deviation from a single Debye-type behavior as observed in this study. Then, we did not use these models to analyze dielectric spectra obtained in this study. However, it is worth noting that the Cole−Davidson model is also quite helpful to describe the broad frequency dependence of the susceptibility of aqueous solutions determined by spectroscopic techniques other than the DS measurement to explore hydration behavior, such as EDLS [27,28,29,30] and NS [33] methods.

Figure 2a,b demonstrate the concentration dependence of the dielectric relaxation times and strength for each mode (*j* = 1 to 5) obtained from the aqueous HeC(1.3:90) solution. The fastest relaxation time, *τ*_1_, was equal to the value of the rotational relaxation time, *τ*_w_, for the pure liquid water molecules, as plotted in Figure 2a. Then, the mode *j* = 1 is easily assigned to the rotational relaxation mode of the free water molecules in the aqueous HeC(1.3:90) solution. The value of *ε*_1_ clearly decreased with increasing concentration, *c*, as shown in Figure 2b. The amount of free water molecules in the aqueous solution was substantially reduced by the volume effect of the solute HeC(1.3:90) molecules, which possess a certain volume fraction in the aqueous solution, and the presence of hydration water. In a later section, the number of hydrated water molecules to a glucopyranose unit of the HeC(1.3:90) molecules (hydration number, *n*_H_) will be calculated from the concentration dependence of *ε*_1_.

The second relaxation mode *j* = 2 possesses the relaxation time *τ*_2_ ~ 2.4*τ*_1_ and the strength *ε*_2_ is the second largest and proportional to *c*, as shown in Figure 2a,b. According to previous research, for several water-soluble substances [23,26,30,35,36,37,38], MD simulation techniques [39,40,41], and DS and EDLS measurements [30], the second mode is assigned to the exchange process of hydrated water molecules by free ones in the aqueous solution. The obtained relaxation time, *τ*_2_, usually possesses a longer relaxation time than the value of *τ*_1_ for some factors. The ratio of *τ*_2_/*τ*_w_ has been called a retardation ratio (or slowdown factor) originally. The evaluated value of *τ*_2_/*τ*_1_ is essentially identical to that of *τ*_2_/*τ*_w_ in this study because the observed value of *τ*_1_ was very close to that of *τ*_w_. Since the ratio of *τ*_2_/*τ*_1_ is ~ 2.4 as seen in Figure 2a and that of other hydroxyethyl cellulose systems examined in this study are quite similar to that reported for aqueous solutions of various saccharides, such as glucose, fructose, sucrose, and glucose oligomers obtained using EDLS and DS techniques [30,36,37,38,45], the hydration properties of aqueous hydroxyethyl cellulose systems are not so different from those of mono- and oligosaccharide molecules. The fact that the second relaxation strength *ε*_2_ is very close to the calculated value of *n*_H_*c*(*ε*_w_/55.6) for the aqueous HeC solution, which is the estimated dielectric relaxation magnitude for hydrated water molecules, assuming that the same relaxation magnitude as free water molecules, powerfully supports the validity of this assignment.

The third relaxation mode *j* = 3 demonstrates the relaxation time *τ*_3_ of 180 ps, as shown in Figure 2a. Although the strength of mode *j* = 3 is considerably smaller than that of mode *j* = 2, the value is proportional to *c*, as seen in Figure 2b. Since HeC(1.3:90) molecules have some polar groups with finite dipole moments, such as hydroxy (−OH) and ether (−O−) groups, the rotational mode of these polar groups would be detected as dielectric relaxation processes. The value of relaxation time (*τ*_3_ ~ 180 ps) for HeC is similar to that for methyl and hydroxypropylmethyl cellulose (*τ*_3_ ~ 220 ps) [24]. Figure 3a shows the *MS* dependence of relaxation strength per one mol of glucopyranose ring, *ε*_3_*c*
^−1^, and the value of *ε*_3_*c*
^−1^ is proportional to *MS*. The fact that the intercept value of *ε*_3_*c*^−1^ ~ 2.5 for HeC is almost the same value of *ε*_3_*c*
^−1^ for pullulan (*ε*_3_*c*^−1^ ~ 2.6), which is a typical water-soluble natural glucan, strongly supports the validity of this assignment. The rotational relaxation time of a hydroxy group would be governed by the lifetime of two kinds of intramolecular hydrogen bonding between two hydroxy groups, and a hydroxy group and an ether group. HeC molecules have two types of ether groups: (i) a native ether group (n-ether group) that cellulose possesses natively and (ii) an additional ether group (ad-ether group) which is introduced by the substitution. The rotational relaxation time of n-ether groups would be longer than that of ad-ether groups because the rotational motion of n-ether groups requires the rotation of the glucopyranose rings of the backbone chains. On the other hand, the ad-ether groups can rotate without the rotation of the backbone chains. Then, *τ*_3_ can be assigned to the rotational relaxation time of hydroxy groups and ad-ether groups controlled by the lifetime of intramolecular hydrogen bonding.

The relaxation modes, *j* = 4 and 5, possess the relaxation times of *τ*_4_ ~ 3.5 and *τ*_5_ ~ 24 ns, respectively, for HeC(1.3:90). The relaxation strength of these modes, *ε*_4_ and *ε*_5_, are proportional to *c* as *ε*_2_ and *ε*_3_, as shown in Figure 2b. Figure 3b represents the *c* dependence of the strength, *ε*_4_ and *ε*_5_, for all the HeC samples. This figure reveals that the magnitudes, *ε*_4_ and *ε*_5_, are simply proportional to *c* and depend on neither *M*_w_ nor *MS* for the HeC samples. Then, the dielectric strength would be controlled only by the dipole moments of the n-ether groups in glucopyranose rings and β-1,4 linkages, not by the dipole moments of ad-ether groups attached to the glucopyranose rings. The relaxation times of the modes, *τ*_4_ and *τ*_5_, for HeC samples were less dependent on both the *MS* and *M*_w_ values. These observations suggest that the modes, *j* = 4 and 5, are attributed to the local motions of glucopyranose rings in HeC molecules dissolved into water. Since the goal of this study is a fundamental argument about the hydration/dehydration behavior of hydroxyethyl cellulose samples in aqueous solution, a detailed discussion on the relaxation modes, *j* = 4 and 5, is not developed here.

### 2.2. Temperature Dependence of Relaxation Times

Figure 4 represents the reciprocal temperature, *T*^−1^, dependence of dielectric relaxation times, *τ*_1_ and *τ*_2_, and the retardation ratio, *τ*_2_/*τ*_1_, for an aqueous solution of HeC(1.3:90) at *c* = 0.23 M as a typical example. The activation energy of the mode *j* = 1 was calculated to be 17.1 kJ mol^−1^ from the slope of the *τ*_1_ data shown in Figure 4. The obtained value ensures the validity of assignment for the mode *j* = 1 because this value is close to that of the rotational relaxation time, *τ*_w_, of water molecules in the pure liquid state. The activation energy of the mode *j* = 2 was evaluated to be 17.8 kJ mol^−1^, which is slightly larger than that of mode *j* = 1. This small difference in the activation energies is because the interaction hydration energy between a hydrated water molecule and a hydration site of the HeC(1.3:90) molecules is slightly larger than that between pure liquid water molecules. Similar temperature dependencies of the *τ*_1_ and *τ*_2_ values were also obtained for aqueous solutions of HeC(2.0:220) and HeC(3.6:70).

The *τ*_2_/*τ*_1_ value is approximately kept at 2.2–2.4 for HeC(1.3:90) and other HeC samples in the temperature range examined. This retardation ratio reasonably agrees with the previously reported values for hydrated water molecules in aqueous saccharide solutions determined at room temperature using EDLS and DS measurements [30,36,37,38], and aqueous solutions of MC and HpMC in the same temperature range as in this study using DS techniques [24]. In the case of HpMC in aqueous solution, the *T*^−1^ dependence of the *τ*_2_/*τ*_1_ value showed a break point at 30 °C with increasing *T* [24]. However, the *T*^−1^ dependence of the *τ*_2_/*τ*_1_ value for the aqueous HeC systems shows no break point in the measured *T* range. A difference in the presence of break point between HeC and HpMC would be related to the difference in the dehydration behavior of the systems.

The retardation ratio of hydrated water molecules for aqueous green fluorescent protein system was reported at two different temperatures, 7 and 30 °C, by using NS techniques. Although the retardation ratio was independent of *T*, the ratio was ranged from 3 to 7 depending on the scattering vector [33]. The retardation ratios of hydrated water molecules for other proteins like lysozyme and oligopeptides were also reported to be 3 to 7 using various methods such as EDLS [30], DS [30,46], and NMR [31,32] measurements, and MD simulations [40,41]. It seems that amide groups forming oligo peptides and proteins have considerably larger retardation ratios of hydrated water molecules than hydroxy and ether groups constructing polysaccharides, as described above and in the previous study of MC and HpMC [24].

### 2.3. Hydration Behavior

From the *c* dependence of *ε*_1_, the hydration number, *n*_H_, per glucopyranose unit can be determined by using the following equation (Equation (2)) [23,26,35]:(2)$$\frac{\varepsilon_{1}}{\varepsilon_{w}}=\frac{1-10^{-3}V_{m}c}{1+10^{-3}V_{m}c/2}-10^{-3}V_{w}cn_{H}$$
where *V*_m_ and *V*_w_ are the partial molar volumes of glucopyranose units and water molecules, respectively, at each measuring temperature in the unit of cm^3^ mol^−1^. The *V*_m_ value was evaluated from the density data of the examined aqueous HeC solution, and the precise values of *V*_w_ are available in the literature. The first term of Equation (2) represents the contribution of the volume excluded by the presence of solute HeC molecules, and the second one represents the hydration effect of the solute molecules, HeC, in this study.

Figure 5 shows the concentration, *c*, dependence of the *ε*_1_/*ε*_w_ values for aqueous solution of HeC(1.3:90) at 20 °C as a typical example. The broken line in the figure, which is theoretically calculated via Equation (2) assuming *n*_H_ = 0, is far from the experimentally obtained *ε*_1_/*ε*_w_ data points. However, the agreement between the data and the solid line, calculated assuming *n*_H_ = 14, is perfect over the entire *c* range examined. Consequently, we concluded that the *n*_H_ value of HeC(1.3:90) at 20 °C is 14. According to the same procedure, the values of *n*_H_ were determined successfully as a function of *T* from 10 to 70 °C for all the aqueous HeC solutions examined.

Figure 6 shows the dependence of *n*_H_ on temperature, *T*, for aqueous HeC(1.3:90) solutions obtained in a concentration range from 0.14 to 0.23 M. The *n*_H_ values determined from aqueous solutions of HeC at different concentrations demonstrated the same value at each examined temperature. Then, one might conclude that the *n*_H_ value is successfully evaluated and has very weak *c* dependence in the *c* range examined. Furthermore, the *n*_H_ value is ca. 15 at 10 °C, and gently reduces to ca. 10 at 70 °C with increasing temperature. The HeC sample possessed water solubility even in a higher temperature range and did not demonstrate a cloud point and gelation in the examined *T* range, as previously reported [18].

Figure 7a shows the comparison of temperature, *T**,* dependencies of *n*_H_ for aqueous solutions of HeC(1.3:90), MC(1.8:95), which has the degree of substitution, *DS*, by methyl groups, Me, of 1.8 and *M*_w_/10^3^ of 95; HpMC(0.15:1.8:75), which has the *MS* by hydroxypropyl groups of 0.15, the *DS* by Me of 1.8 and *M*_w_/10^3^ of 75; and HeMC(0.20:1.5:300), which has the *MS* by hydroxyethyl groups of 0.20, *DS* by Me of 1.5 and *M*_w_/10^3^ of 300. The hydration behavior of MC, HpMC, and HeMC has been reported already [24]. The *n*_H_ data shown in this figure are almost independent of the concentrations of solute cellulose derivatives. In a temperature range lower than 25 °C, the *n*_H_ value of HeC(1.3:90), which possesses the lowest total substitution quantity, is the highest in the four examined cellulose ether samples. Figure 7b represents temperature, *T*, dependence of the hydration number (*N*_H_) per hydrophilic group—hydroxy (−OH) and ether (−O−) groups—for examining the cellulose ether samples in aqueous solution. Although the *n*_H_ value of HeC(1.3:90) is the highest in the four cellulose ether samples, the *N*_H_ values of these cellulose ethers at a low *T* of 10 °C gather into a similar value of ca. 2.5, irrespective of sample species. The number of ether groups for HeC increases with increasing *MS* by hydroxyethyl, He, groups. However, the number of hydroxy groups per glucopyranose unit for HeC is kept at the original value of three for natural cellulose with the increase in the *MS* value by He groups. Then, the reason why HeC(1.3:90) demonstrates the highest *n*_H_ value in the four cellulose ethers is that the average number of hydrophilic groups per glucopyranose unit for HeC(1.3:90) is 6.3. This value is larger than that for other cellulose ethers, such as 5.0 for MC(1.8:95), 5.15 for HpMC(0.15:1.8:75), and 5.2 for HeMC(0.20:1.5:300). Koda et al. [20] reported that the water retention capacity of HeC is higher than that of MC by using the compression method and differential scanning calorimetry measurements. The higher *n*_H_ values of HeC would correspond to the higher water retention capacity of HeC than that of MC.

The decreasing coefficient of hydration number, (∂*n*_H_/∂*T*)*_c_*, for HeC(1.3:90) with increasing *T* is clearly lower than that for other three cellulose ethers, as seen in Figure 7a. The HeC(1.3:90) keeps a high *n*_H_ value, even in a high temperature range (*n*_H_ ~ 10 at *T* = 70 °C), and the *n*_H_ value of HeC(1.3:90) is always much higher than that of other three cellulose ethers in the examined *T* range. In the previous study, the critical hydration number for MC, HpMC, and HeMC samples necessary to be dissolved in water was evaluated to be ca. 5 [24]. In Figure 7b, the *N*_H_ values of MC(1.8:95) and HpMC(0.15:1.8:75) just below the LCST and that of HeMC(0.20:1.5:300) just below the gelation point are ca. 1. Therefore, the critical hydration number for MC, HpMC, and HeMC samples can be described to be ca. 1 per hydrophilic group. The value of *N*_H_ for HeC(1.3:90) is ca. 1.7, even at *T* = 70 °C, as shown in Figure 7b and is much higher than the critical hydration number per hydrophilic group. This is the reason why HeC(1.3:90) can keep high solubility in water even in a high *T* range.

Figure 8a shows temperature, *T*, dependence of hydration number, *n*_H_, per glucopyranose unit for aqueous solutions of HeC(1.3:90), HeC(2.0:220), and HeC(3.6:70), which possess different *MS* by He groups. All the HeC samples examined in this study keep high water solubility over a wide *T* range and never demonstrate a cloud point even in a high *T* range. As the *MS* by He groups increases, the values of *n*_H_ for HeC increase. This is because the number of hydrophilic ether groups per glucopyranose unit increases with increasing *MS* by He groups, and the number of hydroxy groups per glucopyranose unit is kept at three for natural cellulose. The increase in *n*_H_ value resulting from the additional substitution by He groups is more remarkable in a low *T* range; the increase coefficient of the *n*_H_ value per *MS* by He groups at 10 °C is evaluated to be (∂*n*_H_/∂*MS*)*_c_* = 3.3. The effect on the *n*_H_ value by an increase in *MS* in a high *T* range looks weaker than that in a low *T* range; the increase coefficient at 70 °C is 2.2. Then, the *T* dependence of *n*_H_ for HeC is substantially affected by the *MS* value.

The temperature, *T*, dependence of hydration number, *n*_H_, for triethylene glycol (TEG) was also investigated as a model molecule for the substitution groups of HeC. Scheme 2 represents the chemical structure of TEG. Since the structure of TEG is similar to a possible side chain of HeC molecules made from three hydroxyethyl groups, TEG is a reasonable model to compare the hydration behavior of the substitution groups and that of HeC. The *T* dependence of *n*_H_ per molecule of TEG is also shown in Figure 8a. The dehydration behavior of all the HeC samples and TEG with increasing *T* is gentle, as seen in this figure. The reason why the *T* dependence of *n*_H_ for HeC samples becomes greater with an increasing *MS* value, as mentioned above, would be the hydration characteristics of their substitution groups becoming remarkable with increasing *MS*. Figure 8b shows the *T* dependence of hydration number, *N*_H_, per hydrophilic group for the HeC samples and TEG. The dehydration coefficients of *N*_H_ for the four samples with increasing *T* are very similar to each other. This consideration reveals that (∂*N*_H_/∂*MS*)*_c_* ~ 0 irrespective of *T*, although (∂*n*_H_/∂*MS*)*_c_* depends on *T*, as discussed above. Then, we might conclude that the hydration/dehydration behavior for the HeC samples is essentially controlled by the substitution groups like TEG.

Consequently, the decreasing coefficient of *n*_H_ for HeC with increasing *T* is substantially weaker than that for other cellulose ethers examined previously [24], because the substitution groups of HeC do not show significant but gentle dehydration behavior with increasing *T* and determine the hydration/dehydration behavior of the HeC molecules. That is why the HeC samples in aqueous solution possess high water solubility over a wide *T* range and never demonstrate a cloud point and gelation behavior, even in a high *T* range as the MC, HpMC, and HeMC samples in aqueous solution.

## 3. Materials and Methods

### 3.1. Materials

Hydroxyethyl cellulose, HeC, samples used in this study were kindly supplied by Daicel Corporation (Osaka, Japan), and Dow Chemical Japan Limited (Tokyo, Japan). Table 1 summarizes the characteristics of HeC samples investigated in this study. The molecular weight distribution indexes given by a ratio of *M*_w_ to the number-average molecular weight (*M*_n_) were obtained by using size-exclusion chromatographic methods and in a range from 2 to 3. All HeC samples were purified by dialysis and freeze-dried. Triethylene glycol, TEG (>98%), was purchased from Tokyo Chemical Industry Co., Ltd. (Tokyo, Japan). TEG was used as a model compound of a trimer of ethylene oxide, i.e., a possible side chain of HeC molecules made from hydroxyethyl groups.

Highly deionized water with specific resistance higher than 18 MΩ·cm, generated by a Direct-Q 3UV system (Merck Millipore, Darmstadt, Germany), was used as a solvent. The concentrations of cellulose ethers ranged from 0.08 to 0.23 M in glucopyranose units.

### 3.2. Dielectric Spectroscopic Measurements

Two systems were used to measure the dielectric relaxation behavior for aqueous solutions of the HeC samples over a frequency range from 1 M to 50 GHz. A dielectric probe kit, 8507E, equipped with a network analyzer, N5230C, an ECal module N4693A, and a performance probe 05 (Agilent Technologies, Santa Clara, CA, USA), was used for dielectric spectroscopic measurements over a frequency range from 50 M to 50 GHz (3.14 × 10^8^–3.14 × 10^11^ s^−1^ in angular frequency, *ω*). A three-point calibration procedure using *n*-hexane, 3-pentanone, and water as the standard materials was conducted prior to all the DS measurements at each measuring temperature. The details of the three-point calibration procedure used in this study have been described elsewhere [25]. In these systems, the real and imaginary parts (*ε*′ and *ε*″) of the electric permittivity were automatically calculated from the reflection coefficients measured by the network analyzer via a program supplied by Agilent Technologies.

In the lower frequency range from 1 M to 3 GHz (6.28 × 10^6^–1.88 × 10^10^ s^−1^ in *ω*), an RF LCR meter 4287A (Agilent) equipped with a homemade electrode cell with a vacant electric capacity of *C*_0_ = 0.23 pF was used. Open (air), short (0 Ω, a gold plate with the thickness of 0.5 mm), and load (water) calibration procedures were performed prior to sample measurements at each measuring temperature. Dielectric measurements were performed at temperatures ranging from *T* = 10 to 70 °C with an accuracy of ±0.1 °C using a temperature controlling unit made of a Peltier device. The *ε*′ and *ε*″ values were calculated using the relationship *ε*′ = *CC*_0_^−1^ and *ε*″ = (*G* − *G*_DC_)(*C*_0_*ω*)^−1^, where *C*, *G*, and *G*_DC_ represent measured electric capacitance, conductance, and direct current conductance of the sample liquid, respectively. The contribution of *G*_DC_ to the *ε*″ value was adequately removed in a lower *ω* range.

### 3.3. Density Measurements

Density measurements for the sample solutions were conducted using a digital density meter, DMA4500 (Anton Paar, Graz, Austria), to determine the partial molar volumes of the HeC and TEG samples at the same temperature as the performed DS measurements. The uncertainties of measured density and temperature were less than 5.0 × 10^−^^5^ g cm^−^^3^ and 0.03 °C, respectively.

## 4. Conclusions

The temperature dependence of hydration number for hydroxyethyl cellulose, HeC, over a temperature range from 10 to 70 °C was determined exactly by using extremely high-frequency dielectric spectroscopic techniques. All the HeC samples examined in this study, which possess different molar substitution numbers, *MS*, per glucopyranose unit by hydroxyethyl groups, 1.3, 2.0, and 3.6, well dissolve in water over the temperature range examined.

The HeC molecules showed much gentler dehydration behavior with increasing temperature than that of methyl and hydroxypropylmethyl cellulose, MC and HpMC, molecules which demonstrate cloud points clearly at high temperatures. The HeC samples kept a higher hydration number, *N*_H_, per hydrophilic group, hydroxy and ether groups, than the critical *N*_H_ value of ca. 1, even in a high temperature range. The high *N*_H_ values of HeC molecules over a wide temperature range are the reason for their high water-solubility in the temperature range.

The three HeC samples and a model compound, triethylene glycol, for their substitution groups, demonstrated a very similar decreasing coefficient of *N*_H_, (∂*N*_H_/∂*T*)*_c_*, with increasing *T*. The hydration/dehydration behavior of the HeC molecules was essentially controlled by the substitution groups of HeC.

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
