# Peer review of "Hydration/Dehydration Behavior of Hydroxyethyl Cellulose Ether in Aqueous Solution"

_molecules, 2020, doi:10.3390/molecules25204726_

Round 1
Reviewer 1 Report
The authors present a manuscript detailing the hydration behavior of hydroxyethyl cellulose using high frequency dielectric techniques. The manuscript is well written and readily understood. Overall the presented research could be valuable to other researchers in the field. The researchers were very careful in describing the experimental part. Therefore, these experiments could be replicated by other researchers. The conclusions in this manuscript are supported by the experimental results. Thus I support its publication.
Author Response
Thank you so much for your kind comments.
Reviewer 2 Report
In this study, in order to clarify the reason for the high solubility of HeC, the temperature dependence of the hydration number per glucopyranose unit, nH, for the HeC samples was examined by using extremely high frequency dielectric spectrum measuring techniques up to 50 GHz over a temperature range from 10 to 70 ºC. This study strongly suggests that the hydration/dehydration behavior of the HeC samples was essentially controlled by that of their substitution groups. The experiments were well designed and carefully performed, and the manuscript is well organized. Therefore, this manuscript could be considered for publication in Molecules after a minor revision.
- Line 34, insolubility? solubility?
- Line 113, 271and 311: There is an error in the title format, please correct it.
- Line 135-138, the uncertainty of density and temperature shoud be provided.
- Line 425:Please analyze the reason for this phenomenon: "The effect on the nH value by an increase of MS in a high T range looks weaker than that in a low T range."
- Figures in Schemes 1 and 2 are too large.
- The format of multiple references needs to be modified, for example: 1, 4, 15, 39, 40.
- Some important references should be cited such as ACS Omega,2020, 5, 38, 24256-24261; Green Chem., 2019, 21, 4449–445; Coatings, 2020, 10(5), 499; https://doi.org/10.3390/coatings10050499
.
Author Response
Responses to Reviewer 2
Thank you so much for your kind instructive comments and suggestions to our original manuscript, molecules-946363, entitled “Hydration/Dehydration Behavior of Hydroxyethyl Cellulose Ether in Aqueous Solution” by Toshiyuki Shikata and Kengo Arai.
We would like to response to you as follows.
- Insolubility? or solubility?
We would like to use “insolubility” also in the revised manuscript.
- There is an error in the title format, please correct it.
We corrected the title format in the revised manuscript.
- The uncertainty of density and temperature should be provided.
According to you, a sentence below was added in the revised manuscript in Line 466.
“The uncertainties of measured density and temperature were less than 5.0 × 10-5 g cm-3 and 0.03 °C, respectively.”
- Please analyze the reason for this phenomenon: "The effect on the nH value by an increase of MS in a high T range looks weaker than that in a low T range."
As a response to your kind instructive request, we added a sentence below in the revised manuscript in Line 417.
“This consideration reveals that (∂NH/∂MS)c ~ 0 irrespective of T, although (∂nH/∂MS)c depends on T as discussed above.”
- Figures in Schemes 1 and 2 are too large.
According to you, we would like to diminish the sizes of Scheme 1 and 2 in the revised manuscript.
- The format of multiple references needs to be modified.
In accordance with the format of Molecules, we corrected the expression format of some references in the revised manuscript.
- Some important references should be cited such as ACS Omega,2020, 5, 38, 24256-24261; Green Chem., 2019, 21, 4449–445; Coatings, 2020, 10(5), 499.
According to you, we would like to add these suggested references in the revised manuscript as references: 5, 6 and 7.
We would be pleased, if the revised version will be accepted soon.
Reviewer 3 Report
1. The first abbreviation should be defined, for example: Line 51 – LSCT. Please check all.
2. I would like to ask you, to emphasize why these high temperature studies are important and what is the knowledge acquired which can be applied on the mentioned applications (line 48-49)?
3. I would like to ask the authors if it is possible present a schematically the temperature dependent structural modification mechanism for the following cases:
- modification of HeC samples in aqueous solution over a temperature range from 10 to 70 ºC
- influence of the substitution groups with increasing temperature for the HeC, MC, HpMC and HeMC samples
- Influence of the molar substitution number (MS) by using HeC samples with the different MS quantities per glucopyranose unit by hydroxyethyl groups ranged from 1.3 to 3.6.
Author Response
Responses to Reviewer 3
Thank you so much for your kind instructive comments and suggestions to our original manuscript, molecules-946363, entitled “Hydration/Dehydration Behavior of Hydroxyethyl Cellulose Ether in Aqueous Solution” by Toshiyuki Shikata and Kengo Arai.
We would like to response to you as follows.
- The first abbreviation should be defined, for example. Please check all.
According to you, we added some necessary definitions for abbreviations in the revised manuscript.
- I would like to ask you, to emphasize why these high temperature studies are important and what is the knowledge acquired which can be applied on the mentioned applications?
According to your kind suggestion, we added a sentence below in the revised manuscript in Line 89.
“It is important to understand why chemically modified cellulose samples possess rather different temperature dependencies of hydration behavior and solubility highly depending on the species of substitution groups for their many advanced practical applications especially at high temperatures.”
- I would like to ask the authors if it is possible present a schematically the temperature dependent structural modification mechanism for the following cases:
- i) Modification of HeC samples in aqueous solution over a temperature range from 10 to 70 ºC.
- ii) Influence of the substitution groups with increasing temperature for the HeC, MC, HpMC and HeMC samples.
iii) Influence of the molar substitution number (MS) by using HeC samples with the different MS quantities per glucopyranose unit by hydroxyethyl groups ranged from 1.3
These requests given by you are not easy for us at this stage. However, we only attached a quite simple schematic depiction to show a difference in the temperate dependencies of hydration/dehydration behaviors for HeC and MC in aqueous solution as a graphic abstract, GA, to the revised manuscript, which is related to the point of ii).
We would be pleased, if the revised version will be accepted soon.